# Optical neural engine for solving scientific partial differential equations

Yingheng Tang [1,4] ✉, Ruiyang Chen [2,4], Minhan Lou [2], Jichao Fan[2], Cunxi Yu [3], Andrew Nonaka[1], Zhi Yao [1] ✉ & Weilu Gao [2] ✉

Solving partial differential equations (PDEs) is the cornerstone of scientific research and development. Data-driven machine learning (ML) approaches are emerging to accelerate time-consuming and computation-intensive numerical simulations of PDEs. Although optical systems offer high-throughput and energy-efficient ML hardware, their demonstration for solving PDEs is limited. Here, we present an optical neural engine (ONE) architecture combining diffractive optical neural networks for Fourier space processing and optical crossbar structures for real space processing to solve time-dependent and time-independent PDEs in diverse disciplines, including Darcy flow equation, the magnetostatic Poisson's equation in demagnetization, the Navier-Stokes equation in incompressible fluid, Maxwell's equations in nanophotonic metasurfaces, and coupled PDEs in a multiphysics system. We numerically and experimentally demonstrate the capability of the ONE architecture, which not only leverages the advantages of high-performance dual-space processing for outperforming traditional PDE solvers and being comparable with state-of-the-art ML models but also can be implemented using optical computing hardware with unique features of low-energy and highly parallel constant-time processing irrespective of model scales and real-time reconfigurability for tackling multiple tasks with the same architecture. The demonstrated architecture offers a versatile and powerful platform for large-scale scientific and engineering computations.

Partial differential equations (PDEs) derived from physical laws have been a powerful and faithful computational tool to accelerate the exploration and validation of scientific hypotheses instead of performing expensive and time-consuming real-world experiments[1]. Hence, numerically solving PDEs is essential for scientific research and development in nearly every scientific domain. For example, the interaction of electromagnetic waves with materials and engineered structures in broad applications such as communication, imaging, sensing, and quantum technologies is governed by Maxwell's equations[2]; automotive and flight aerodynamics for designing and manufacturing road vehicles and airplanes is determined by Navier-

Stokes equations[3]; the Earth system including temperature, atmosphere, and ice sheets for understanding climate change and making policies is also described with a series of PDEs[4]. However, current numerical simulation methods to solve PDEs, such as finite difference/volume methods to solve Maxwell's and Navier-Stokes equations, are costly in computing time and resources.

Machine learning (ML) offers a new perspective on solving PDEs through data-driven approaches to enable fast and accurate simulations of many multiphysics and multiscale processes[5–7]. However, the ML model deployment on electronic computing hardware requires substantial computing resources and consumes substantial energy. In

[1]Center for Computational Sciences and Engineering, Lawrence Berkeley National Laboratory, Berkeley, CA, USA. [2]Department of Electrical and Computer Engineering, The University of Utah, Salt Lake City, UT, USA. [3]Department of Electrical and Computer Engineering, University of Maryland, College Park, MD, USA. [4]These authors contributed equally: Yingheng Tang, Ruiyang Chen. ✉e-mail: ytang4@lbl.gov; jackie_zhiyao@lbl.gov; weilu.gao@utah.edu

the foreseeable future, the fundamental quantum mechanics limit will lead to a bottleneck of further reducing the energy consumption and simultaneously increasing the integration density of electronic circuits to catch up with the increasing scale of ML models in demand for solving complex problems[8,9], thus urgently calling for new high-throughput and energy-efficient ML hardware accelerators. Recently, optical architectures, including photonic integrated circuits for matrix-vector multiplication (MVM)[10,11], for neuro-inspired spiking neural networks[12,13], and for photonic reservoir computing[14,15], and free-space optical systems for MVM[16–18] and diffractive optical neural networks (DONNs)[19–22], are emerging as high-performance ML hardware accelerators by leveraging different particles – photons – to break down electronic bottleneck thanks to high parallelism and low static energy consumption of photons[23]. However, the deployment of optical computing systems is in small scales for basic PDEs with limited performance[24,25].

Here, we present a fully reconfigurable and scalable optical neural engine (ONE) architecture that combines DONN systems for processing data in Fourier space and optical crossbar (XBAR) structures for processing data in real space to solve two-dimensional (2D) spatiotemporal profiles in time-independent and time-dependent PDEs. The ONE architecture not only leverages the advantages of high-performance dual-space processing[26], but also can be implemented using optical computing hardware with unique features of low-energy and highly parallel constant-time processing irrespective of model scales, and real-time reconfigurability for tackling multiple tasks with the same architecture. We numerically and experimentally demonstrate the capability of the ONE architecture in solving a broad range of PDEs in diverse disciplines, including the Darcy flow equation in fluid dynamics, the magnetostatic Poisson's equation in micromagnetics, the Navier-Stokes equation in aerodynamics, Maxwell's equations in nanophotonics, and coupled electric current and heat transfer equations in a multiphysics electrical heating problem. The ONE architecture not only outperforms traditional PDE solvers because of its data-driven nature, but also shows comparable and better performance with other ML models while with substantial hardware advantages because of its implementation in the optical domain. The demonstrated ONE architecture is versatile and can be tailored with different combinations of DONN and XBAR structures for solving various PDEs, offering a transformative universal solution for large-scale scientific and engineering computations.

## Results

### ONE architecture

Figure 1a illustrates the ONE architecture, which takes the spatiotemporal data of an input physical quantity $U$, described as a function $u(x, y, t)$ in terms of positions $x$ and $y$ and time $t$, to predict the spatiotemporal data of an output physical quantity $G$ described using a function $g(x, y, t)$. The input and output quantities $U$ and $G$ can be connected through either a single-physics PDE or coupled multiphysics PDEs. There are three branches inside the ONE architecture, including (i) Fourier space processing branch, (ii) real space processing branch, and (iii) physics parameter processing branch. The combination of both real and Fourier space processing has been proven fast, powerful, and efficient in solving PDEs[26], and the incorporation of additional physics parameter processing enables the fusion of multimodal data for complex tasks[27]. More importantly, most operations in these branches can be deployed on optical computing hardware in both real and Fourier spaces, enabling solving PDEs in high-throughput and energy-efficient manners. The details of each branch are described below.

In the first Fourier space processing branch, the core arithmetic operations are based on Fourier and inverse Fourier transform to process input spatiotemporal data in the Fourier space. Their optical hardware implementations are mainly based on reconfigurable

DONNs, which contain cascaded reconfigurable diffractive layers. Reconfigurable DONNs can be implemented in both integrated photonic chips[28,29] and free space[19–21]; see Fig. 1b. There are two fundamental operations in DONNs – optical diffraction and spatial light modulation. For the optical diffraction operation, an optical field right after the $l$-th diffractive layer, $f_l$, diffracts to the front of $(l + 1)$-th layer, whose optical field, $f_{in,l+1}$, is a convolution of $f_l$ and the diffraction impulse function $h(x, y)$. Specifically, the complex-valued field at point $(x, y)$ on the input plane of $(l + 1)$-th layer can be written as the convolution of all fields at the output plane of $l$-th layer as

$$f_{in,l+1}(x,y,z) = \iint f_l(x',y',0)h(x-x',y-y')dx'dy', \quad (1)$$

where $z$ is the distance between two diffractive layers and $h(x, y)$ is the impulse response function of free space. By the convolution theorem, this 2D convolution can be efficiently calculated in Fourier space based on Fourier and inverse Fourier transform. Specifically, the 2D Fourier transform $\mathcal{F}_{xy}$ of $f$ and $h$, $F$ and $H$, are connected through

$$\mathcal{F}_{xy}(f_{in,l+1}(x,y,z)) = \mathcal{F}_{xy}(f_l(x,y,0))\mathcal{F}_{xy}(h(x,y)), \quad (2)$$

$$F_{in,l+1}(\alpha,\beta,z) = F_l(\alpha,\beta,0)H(\alpha,\beta), \quad (3)$$

where $\alpha$, $\beta$ are spatial domain indices. After diffraction, the 2D inverse Fourier transform $\mathcal{F}_{xy}^{-1}$ of $F_{in,l+1}(\alpha, \beta, z)$, $f_{in,l+1}(x, y, z)$, is then spatially modulated. Each diffraction pixel at location $(x, y)$ has a complex-valued electric field transmission coefficient $t(x, y, S)e^{\phi(x, y, S)}$, where $t(x, y, S)$ ($\phi(x, y, S)$) is the amplitude (phase) response as a function of external stimuli $S$, such as voltages. The spatial light modulation operation is expressed as a pixel-wise multiplication

$$f_{l+1}(x,y,z) = \mathcal{F}_{xy}^{-1}(F_{in,l+1}(\alpha,\beta,z))t(x,y,S)e^{\phi(x,y,S)} = f_{in,l+1}(x,y,z)t(x,y,S)e^{\phi(x,y,S)}, \quad (4)$$

where $f_{l+1}(x, y, z)$ is the near-field output field right after the $(l + 1)$-th layer. More details can be found in Methods.

Before and between DONN kernels, there is a linear transformation operation based on fully connected layers to scale up the number of channels and a channel mixing operation based on matrix multiplications[26]. The core arithmetic operations are based on MVM. Their optical hardware implementations are mainly based on reconfigurable optical XBAR structures, which encode element values of vector $\mathbf{v}$ and matrix $M$ into light intensity through electro-optic modulators, perform multiplications through cascaded modulators, and add signals at the output detector array. The signals are routed to follow mathematical calculations in MVM so that the reading from the detector array represents the output vector $\mathbf{o} = M \times \mathbf{v}$. Reconfigurable XBAR structures can also be implemented in both integrated photonic chips[10,11] and free space[16–18]; see Fig. 1c. More details on the operation mechanism can be found in Methods and Supplementary Fig. 1.

The second real space processing branch contains fully connected layers, whose operations are also based on MVM and implemented with optical XBAR structures. The output from the Fourier space branch, $F(u)$, and the output from the real space branch, $R(u)$ are added and further processed with a nonlinear operation. Note that the nonlinear operation is the only operation performed in electronic hardware in the ONE architecture. In practice, fast digital and analog circuits are available to match the speed of optical hardware[30,31] and can be arranged in an array to process each data element in parallel without limiting system throughput. Further, the energy consumption and throughput of nonlinear operations become asymptotically negligible compared to linear operations when the problem scale is large enough (see detailed discussions at the end). Moreover, nonlinear

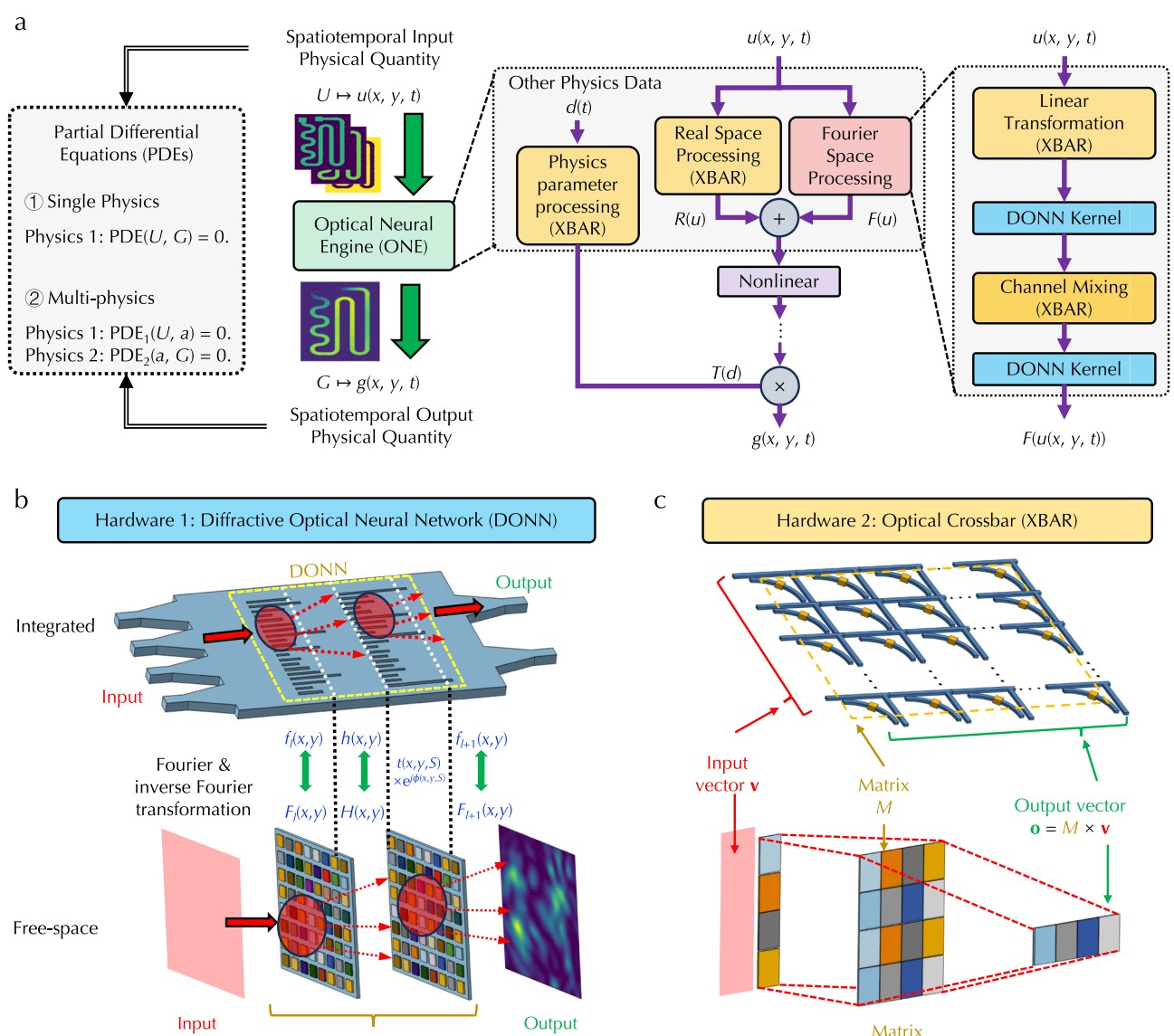

**Fig. 1 | Optical neural engine (ONE) architecture and hardware implementations. a** Illustration of processing branches and flows in the ONE architecture to predict output spatiotemporal output physical quantities from corresponding input and solve partial differential equations involving single or multiple physics. Illustrations of integrated and free-space implementations of reconfigurable (**b**) diffractive optical neural network and (**c**) optical crossbar structures.

processing has been recently achieved even with linear optical systems[32], which could replace electronic hardware nonlinear operations in the future. Moreover, this combination of real space, Fourier space, and nonlinear processing is scaled up, repeated four times, and cascaded in series. The third branch is to perform a linear transformation on other relevant physics parameters $d(t)$, which are time sequences instead of spatiotemporal data, based on fully connected layers. The obtained data $T(d)$ is multiplied and merged onto two other branches to have the final output $g(x, y, t)$. Hence, except for nonlinear operations, all other operations can be done with DONN and optical XBAR systems. These two systems can be seamlessly assembled into a single integrated photonic chip or a single free-space optical system for all-optical operations without converting between optical and electronic hardware, fully leveraging the advantages of high throughput and high parallelism in optical computing systems. More details on the ONE architecture model are in Methods.

### Darcy flow and magnetostatic Poisson's equations
The first PDE we solved with the ONE architecture is the Darcy flow equation in fluid dynamics physics. This PDE describes a fluid flow through a porous medium as shown in Fig. 2a. Specifically, the equation is

$$-\nabla \cdot (k(x,y)\nabla u(x,y)) = f(x,y), \qquad (5)$$

where $k(x, y)$ is the permeability field of the medium, $u(x, y)$ is the pressure field of the flow, and $f(x, y)$ is the force function. The ONE architecture was trained to learn the mapping from the 2D function $k(x, y)$ to function $u(x, y)$. More details about the equation dataset generation and training are in Methods. Figure 2b displays the training loss curves for inputs with different resolutions. The training loss is generally low for all resolutions and slightly increases at the highest 421 resolution. Figure 2c shows the comparison of the training loss of our ONE architecture with other PDE solving models, including fully convolution networks (FCN)[33], principal component analysis-based neural network (PCANN)[34], reduced biased method (RBM)[35], graph neural operator (GNO)[36], low-rank kernel decomposition neural operator (LNO)[27], multipole graph neural operator (MGNO)[37], and Fourier neural operator (FNO)[26]. The performance of the ONE architecture is comparable with state-of-the-art neural operators

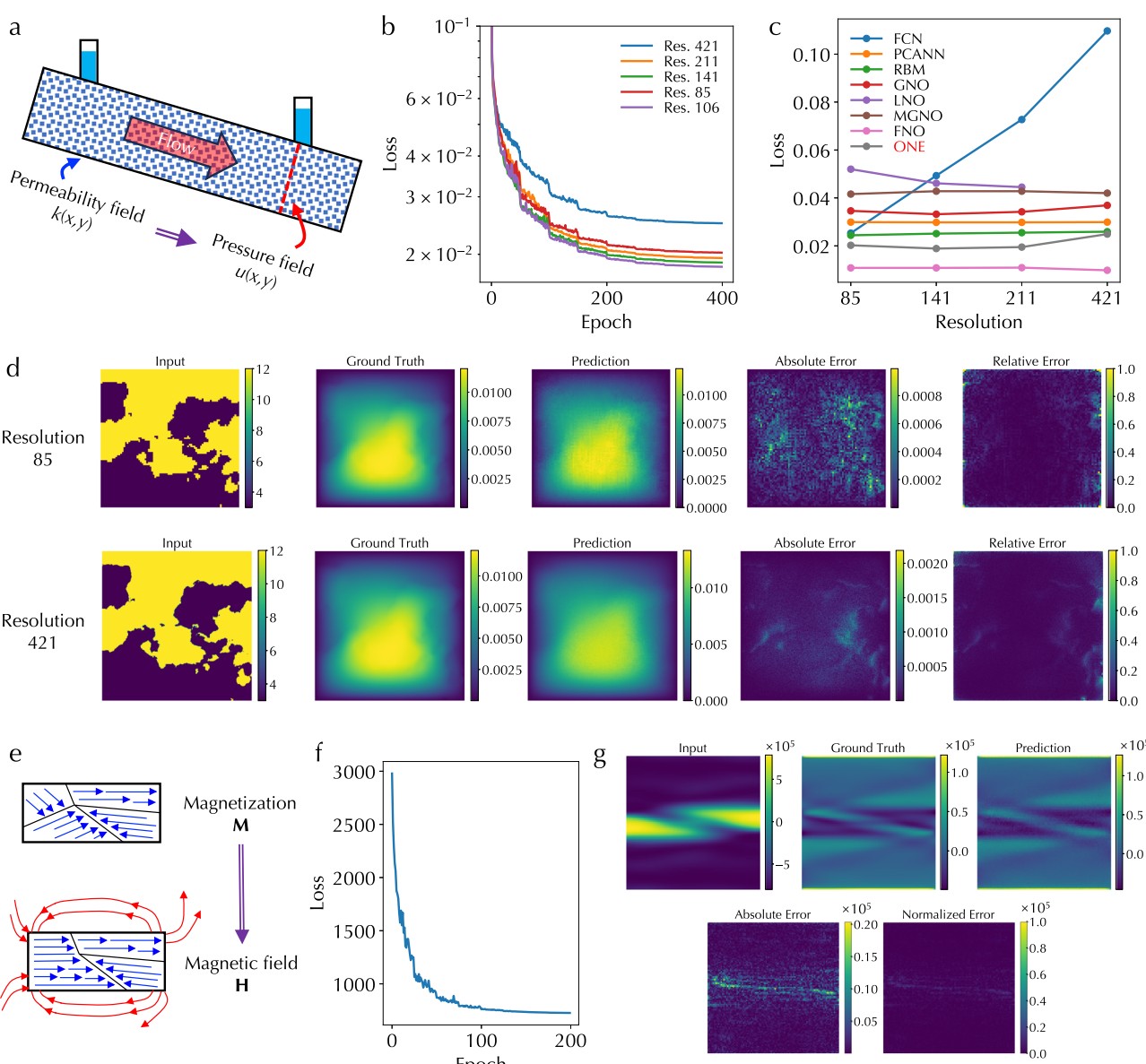

**Fig. 2 | Simulation results of solving Darcy flow and magnetostatic Poisson's equations. a** Illustration of the Darcy flow equation describing a fluid flow through a porous medium. The optical neural engine (ONE) architecture learns the mapping between the permeability and pressure fields. **b** Training loss curves for input data with the input data resolutions of 85, 106, 141, 211, and 421. **c** Comparison of the training loss of fully convolution networks (FCN), principal component analysis-based neural network (PCANN), reduced biased method (RBM), graph neural operator (GNO), low-rank kernel decomposition neural operator (LNO), multipole graph neural operator (MGNO), Fourier neural operator (FNO), and ONE models at various resolutions. **d** Input permeability field, the expected ground truth of output pressure field, the predicted output pressure field, the absolute error between the expected and predicted outputs, and the relative error between the expected and predicted outputs, at 85 and 421 resolutions. **e** Illustration of the magnetostatic Poisson's equation calculating the demagnetizing field generated by the magnetization field. The ONE architecture learns the mapping between these two fields. **f** Validation loss curve for the ONE architecture solving the magnetostatic Poisson's equation and (**g**) corresponding input magnetization field, the expected ground truth of output demagnetizing field, the predicted output demagnetizing field, the absolute and normalized errors between the expected and predicted outputs.

including GNO, LNO, MGNO, and FNO, and is better than FCN. Further, from the hardware perspective, the ONE architecture is constructed based on high-throughput optical computing hardware platforms so that all operations can be performed in parallel and in a single-shot manner. In addition, the ONE architecture can be practically implemented on a large scale. For example, free-space reconfigurable DONNs[20,21,38] and optical MVM[17] are typically implemented using spatial light modulators (SLMs) with a scale >1000×1000. Hence, the execution cost of solving PDEs with different scales and resolutions is invariant, meaning $O(1)$, if the scale of the optical hardware in the ONE

architecture is large enough. Figure 2d displays the input permeability field $k(x, y)$, the expected ground truth of output pressure field $u(x, y)$, the predicted output pressure field, the absolute error between the ground truth and prediction, and the relative error defined as the ratio of the absolute error over the ground truth, at the lowest 85 and the highest 421 resolutions, respectively. This visualization further validates the ONE architecture in solving PDEs. More data on other resolutions are shown in Supplementary Fig. 2.

The second PDE we solved is the magnetostatic Poisson's equation of demagnetization in micromagnetics physics. This PDE

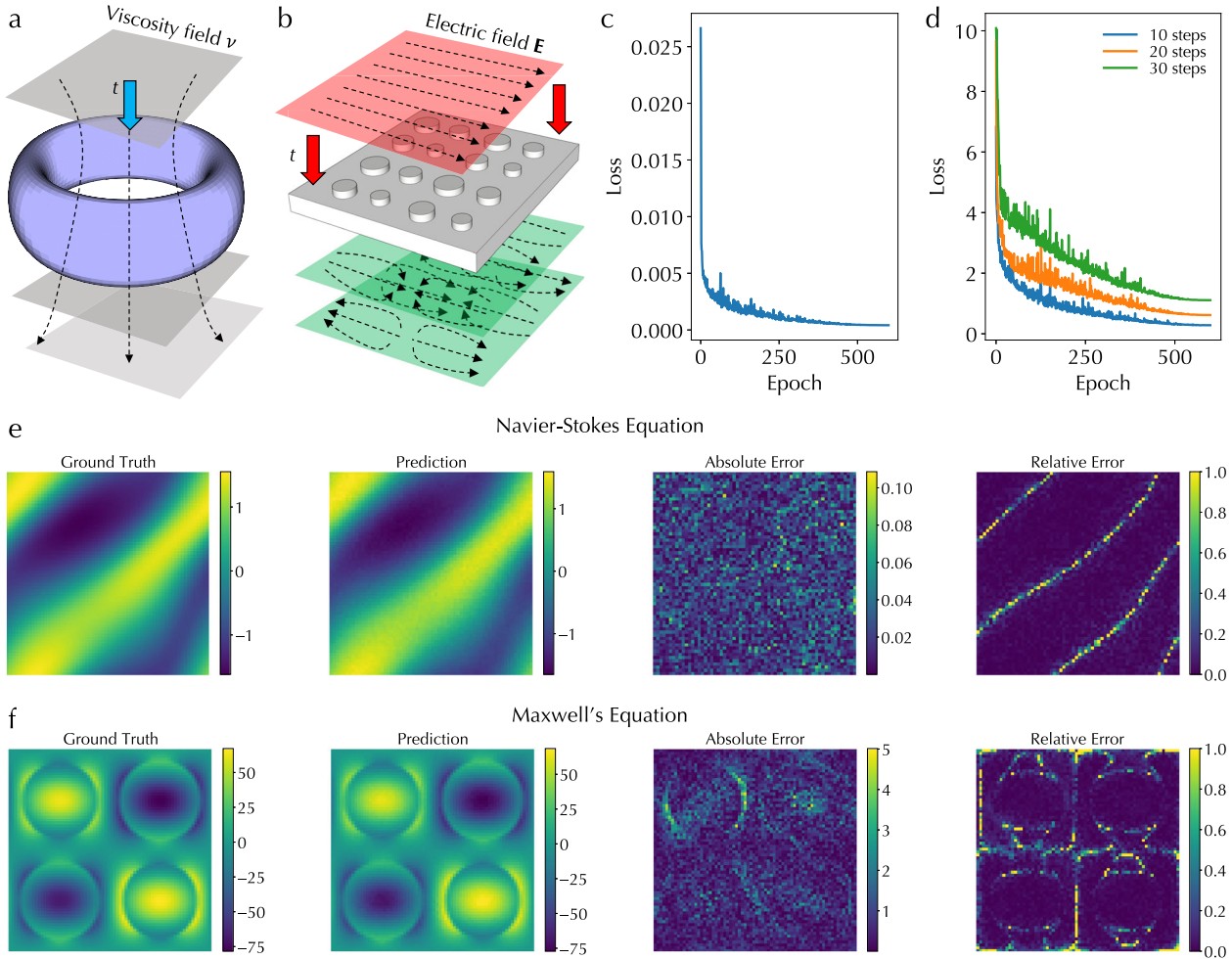

**Fig. 3 | Simulation results of solving time-dependent Navier-Stokes and Maxwell's equations.** Illustrations of (**a**) Navier-Stokes equation for solving the time evolution of the vorticity field in a viscous, incompressible fluid in vorticity form on the unit torus and (**b**) Maxwell's equations for solving the time evolution of the electric field in a dielectric metasurface. Validation loss curves for (**c**) solving the Navier-Stokes equation and (**d**) Maxwell's equations with 10, 20, and 30 additional time steps using the optical neural engine architecture. The expected ground truth field, the predicted field, and the absolute and relative errors between these two fields for (**e**) the Navier-Stokes equation and (**f**) Maxwell's equations, respectively.

calculates the demagnetizing field **H** generated by the magnetization field **M** as shown in Fig. 2e. Specifically, the equation is obtained from Maxwell's equation as

$$\nabla \cdot \mathbf{H} = -\nabla \cdot \mathbf{M}. \tag{6}$$

By defining an effective magnetic charge density $\rho_{mag} = -\nabla \cdot \mathbf{M}$ and a magnetic scalar potential $\psi$ assuming there is no free current, we can express the demagnetizing field $\mathbf{H} = -\nabla\psi$ and rewrite the previous equation as a Poisson's equation

$$\nabla^2\psi = -\rho_{mag}. \tag{7}$$

Similar to solving the Darcy flow equation, the ONE architecture was trained to learn the mapping from components of **M** to **H** vector fields. More details about the equation dataset generation and training are in Methods. Figure 2f shows the validation loss curve and Fig. 2g shows the input one component of **M** field, the expected ground truth of output $H_x$ component of **H** field, the predicted output $H_x$ component, the absolute error between the expected and predicted outputs, and normalized error between the expected and predicted outputs

with respect to the maximum field strength in the ground truth. Both confirm a good performance of the ONE architecture in solving the magnetostatic Poisson's equation. More data on $H_y$ and $H_z$ components are shown in Supplementary Fig. 3.

## Navier-Stokes and Maxwell's equations

In addition to steady-state Darcy flow and magnetostatic Poisson's equations without time evolution, we employed the ONE architecture to solve time-dependent PDEs, including the Navier-Stokes equation in fluid dynamics and Maxwell equations in electromagnetics and optics. In particular, the real-time reconfigurability of DONN and optical XBAR structures makes the ONE architecture suitable for such a purpose. Specifically, we solved a 2D Navier-Stokes equation for a viscous, incompressible fluid in vorticity form on the unit torus as shown in Fig. 3a. This PDE calculates the time evolution of vorticity described as

$$\partial_t w(x,y,t) + u(x,y,t) \cdot \nabla w(x,y,t) = \nu\Delta w(x,y,t) + f(x,y), \tag{8}$$

where $u$ is the velocity field, $w = \nabla \times u$ is the vorticity, $\nu$ is the viscosity coefficient, $f$ is the forcing function. The ONE architecture was trained to learn the mapping from $w$ in a time range from 0 to $t_0$ to $w$ in a time

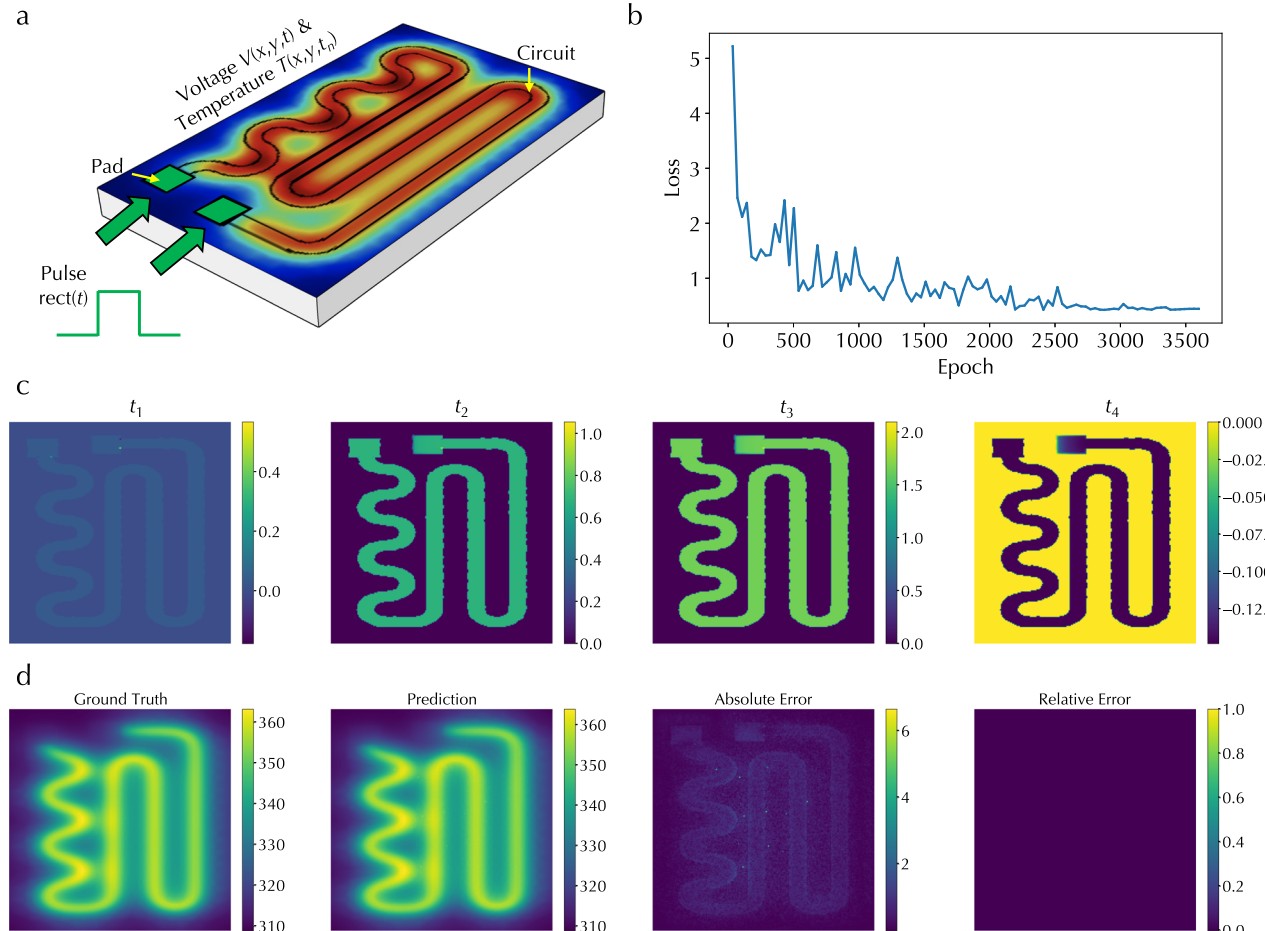

**Fig. 4 | Simulation results of solving multiphysics partial differential equations. a** Illustration of solving coupled PDEs in an electrical heating problem involving electric current physics and heat transfer physics. **b** Validation loss curve. **c** A few representative 2D voltage profiles in the circuit at different times $t_1$, $t_2$, $t_3$, $t_4$. **d** The expected ground truth temperature profile, the predicted profile, and the absolute and relative errors between these two profiles.

range from $t_0$ to $t_1$ ($t_1 > t_0$). More details about the equation dataset generation and training are in Methods. Further, we also solved Maxwell's equations in a dielectric metasurface consisting of multiple cylindrical pillars in a unit cell of a periodic pattern as shown in Fig. 3b[39]. The general Maxwell's equations can calculate the time evolution of an electric field through the following equations

$$\nabla \cdot \mathbf{D} = \rho, \tag{9}$$

$$\nabla \cdot \mathbf{B} = 0, \tag{10}$$

$$\nabla \times \mathbf{E} = -\frac{\partial \mathbf{B}}{\partial t}, \tag{11}$$

$$\nabla \times \mathbf{H} = \mathbf{J} + \frac{\partial \mathbf{D}}{\partial t}, \tag{12}$$

where **D** is the electric displacement field, $\rho$ is the free electric charge density, **B** is the magnetic flux density, **E** is the electric field, **H** is the magnetic field, and **J** is the free current density. The ONE architecture was trained to learn the mapping from **E** in a time range from 0 to $t_0$ to **E** in a time range from $t_0$ to $t_1$ ($t_1 > t_0$). More details about the dataset generation and training are in Methods. Figure 3c displays the validation loss curve for solving the Navier-Stokes equation with $t_0 = 10$ and $t_1 = 20$. Figure 3d displays the validation loss curves for solving

Maxwell's equations with $t_0 = 10$ and $t_1 = 20$, 30, 40, respectively. Moreover, Figure 3e, f show the expected ground truth of $w$ field and the $E_x$ component of the **E** field at $t_1$, the corresponding predicted fields at $t_1$, and the absolute and relative errors between ground truth and prediction for the Navier-Stokes equation and Maxwell's equations, respectively. Note that the sharp lines in the relative error plot of Fig. 3e is due to the division of small ground truth values. All confirm a good performance in solving time-dependent PDEs using the ONE architecture.

## Multiphysics PDEs
Moreover, we employed the ONE architecture to solve coupled PDEs involving two physics. Specifically, we solved an electrical heating problem to obtain a temperature profile at an intermediate time step $t_n$, $T(x, y, t_n)$, in an electrical circuit when a time-dependent voltage signal was applied to the circuit pads, involving coupled electric current physics and heat transfer physics; see Fig. 4a. Specifically, for the electrical current physics, the corresponding PDE is

$$Q = d\sigma \nabla_t V(x, y, t), \tag{13}$$

$$V(x_0, y_0, t) = \text{rect}(t), \tag{14}$$

where $Q$ is the heat rate per unit area from an electromagnetic heating source, $\sigma$ is the conductivity of the heating layer, $d$ is the thickness of

the heating layer, $V(x, y, t)$ is the voltage profile in the circuit that is subjected to a voltage boundary condition defined in the pads $V(x_0, y_0, t)$, and $V(x_0, y_0, t)$ is a pulse rectangular function rect($t$) with pulse height and width. For heat transfer physics, the corresponding PDE is

$$\rho C \frac{\partial T}{\partial t} + \rho C \mathbf{u} \cdot \nabla T - \nabla \cdot (k \nabla T) = Q, \qquad (15)$$

where $\rho$ is the mass density, $C$ is the specific heat capacity, $T$ is the absolute temperature, $\mathbf{u}$ is the velocity of the heated flow, and $k$ is the thermal conductivity. These two PDEs are connected through the quantity $Q$. The ONE architecture was trained to learn the mapping from $V(x, y, t)$ in a time range spanning all time steps in input pulses to $T(x, y, t_n)$ at an intermediate pulse time step $t_n$. In contrast to previous examples, the pulse information, including pulse height and width, was processed through the physics parameter processing branch in the ONE architecture (Fig. 1a) and multiplied with the output from cascaded real space processing and Fourier space processing branches to yield the final output. More details about the dataset generation and training are in Methods. Figure 4b displays the validation loss curve and Fig. 4c shows a few representative input 2D data $V(x, y, t)$ at various time steps. Figure 4d shows the expected ground truth of $T(x, y, t_n)$, the corresponding predicted temperature profile, and the absolute and relative errors between the ground truth and prediction. All confirm a good performance in solving multiphysics PDEs using the ONE architecture.

## Experimental demonstration

Finally, to demonstrate the experimental feasibility of the ONE architecture, we constructed a free-space reconfigurable DONN setup and evaluated the performance of solving the Darcy flow and Navier-Stokes equations. Figure 5a displays a photo and schematic of the reconfigurable DONN setup, which contains a laser source, a reconfigurable input encoder, two reconfigurable diffractive layers, and a camera. The reconfigurable encoder and diffractive layers were built using SLMs, which can modulate the amplitude and phase of transmitted light when applying voltages. Multiple light polarization components, including polarizers and half-wave plates, were also employed to manipulate polarization states to achieve large phase modulation ranges. More details on the experimental setup are in Methods.

As shown in Supplementary Fig. 4, the experimentally measured amplitude and phase modulation responses of all three SLMs are not only discrete with respect to gray levels but also coupled and dependent. To leverage the gradient-based ML training algorithm, we utilized the Gumbel-softmax reparameterization technique to approximate a discrete distribution to a continuous distribution[21]. More details are described in Methods. Moreover, the values of input 2D data span both negative and positive values and were encoded as the gray level of the SLM in the reconfigurable input encoder (SLM0 in Fig. 5a). We performed the encoding through linear mapping from the minimum and maximum values of input data to a gray-level range in the SLM. More details are described in Methods. In addition, we precisely aligned all SLMs with respect to each other within a range of a few pixels on the order of hundreds of $\mu$m; see Supplementary Fig. 5. Although the long optical path in the system makes the alignment sensitive to external variations, the system's full reconfigurability can enable fast adaptive pixel-by-pixel re-alignment.

The first and second rows of Fig. 5b, c show output 2D data in one DONN kernel of the Fourier space processing branch in the ONE architecture (Fig. 1a) obtained from model calculations and experimental measurements in solving the Darcy flow and Navier-Stokes equations, demonstrating good agreement and experimentally validating the feasibility of the ONE architecture. There are some speckles in the background of measured images, which probably originate from

high-order diffraction interference, leading to numerical errors in the ONE architecture for performing regression tasks. This discrepancy between models and experiments can be mitigated through post-processing. Specifically, we trained a lightweight convolutional neural network (CNN) that takes experimental images as input and model results as output; see Methods for more details on the CNN model. As shown in the third row of Fig. 5b, c, the trained CNN can nearly perfectly post-process experimental results to match model results. More data is shown in Supplementary Figs. 6 and 7. In addition, we trained a baseline architecture by replacing DONN kernels in the architecture shown in Fig. 1a with lightweight post-processing CNN models for solving Darcy flow equations under different resolutions and the Navier-Stokes equation (Supplementary Fig. 8). The losses for the CNN baseline architecture are high in both datasets with poor performance when compared to the results obtained from the ONE architecture with DONN kernels, highlighting the necessity and significance of DONN systems and the limited capability of CNN models.

Instead of post-processing, model-experiment discrepancies can be mitigated through the improvement of system hardware and novel training approaches in future implementations. From the hardware perspective, for example, incorporating moving diffusers in optical systems to remove speckles[40] and employing high signal-to-noise ratio cameras can reduce discrepancies. Moreover, from the training perspective, general time-independent model-experiment discrepancies, such as those due to fabrication non-uniformity, can be mitigated using a physics-aware training approach by incorporating loss functions based on experimental results for gradient calculations and hardware reconfiguration, as demonstrated in prior works[20,38,41]. Further, fine-tuning using this approach can be performed periodically to dynamically adjust reconfigurable hardware in the ONE architecture to mitigate time-dependent model-experiment discrepancies from time-dependent system variations, such as the gradual degradation of optical alignment and temperature variations. Hence, the hardware reconfigurability in the ONE architecture is crucial to enabling approaches that can mitigate various practical deployment challenges and improve system robustness.

We also evaluated the performance of the ONE architecture for solving the Darcy flow equation under different noise levels of optical XBAR structures. Note that we did not experimentally construct an XBAR system for the ONE architecture. Instead, we added random Gaussian noise with zero mean and varying standard deviation (Std) to the values obtained from matrix multiplications in models to represent experimental hardware noise, such as shot noise in photodetectors as shown in our prior experimental demonstration of a free-space XBAR system[42]. The corresponding MVM results and histograms of different noise standard deviation values are shown in Supplementary Fig. 9, and more details can be found in Methods. We trained the ONE architecture with noisy XBAR structures, named noise-aware training. As shown in Fig. 5d and the red dashed line in Fig. 5e, the validation loss increases with the increasing noise standard deviation value. The current hardware implementation of optical XBAR structures with advanced components and calibration algorithms[16–18], including the structure we demonstrated before[42], can achieve quite a small noise level similar or below the noise level corresponding to 0.5 Std. Hence, the noise influence in optical XBAR structures on the performance of the ONE architecture is not substantial. Further, we trained the ONE architecture with the ideal XBAR structure without noise, named noise-unaware training, and performed inference under various XBAR noise Std values. As shown in the blue dashed line of Fig. 5e, the validation losses obtained using the noise-unaware training method are much more prominent than those obtained using the noise-aware training method. In addition, we analyzed the propagation of calculation errors by evaluating mean squared errors between 2D data with and without XBAR noises at the end of each processing unit in the ONE architecture (Supplementary Fig. 10a). As shown in Supplementary Fig. 10b, the

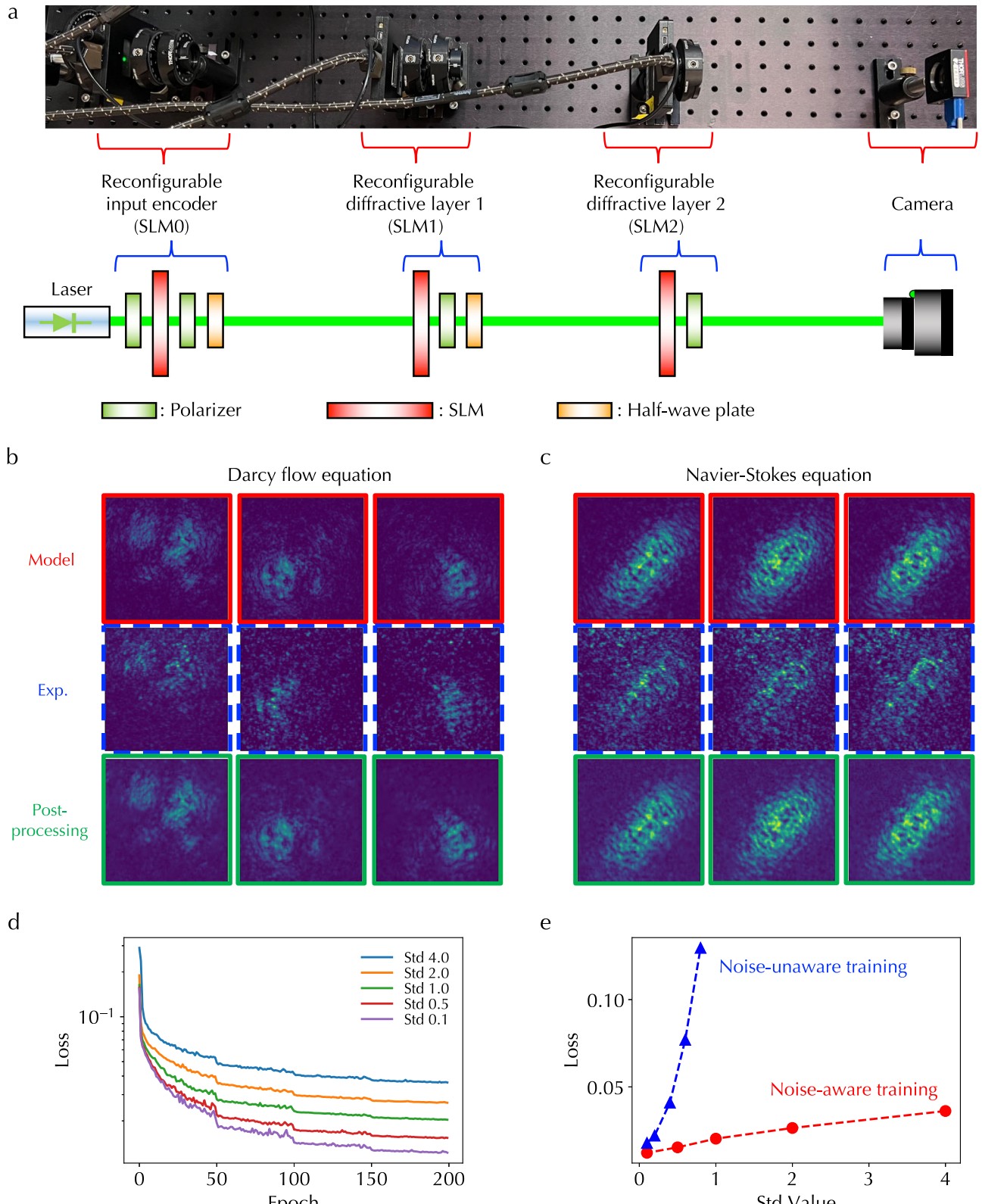

**Fig. 5 | Experimental demonstration. a** Photo and schematic of a reconfigurable diffractive optical neural network (DONN) experimental setup consisting of a reconfigurable input encoder, two reconfigurable diffractive layers, and a camera. Polarization components were used to configure spatial light modulators (SLMs) in the phase modulation mode. Output 2D data in one DONN kernel of the Fourier space processing branch in the optical neural engine architecture obtained from model calculations, experimental measurements (Exp.), and post-processing for solving (**b**) Darcy flow and (**c**) Navier-Stokes equations, respectively. **d** Validation loss curves at different noise levels, represented using 0.1, 0.5, 1.0, 2.0, and 4.0 standard deviation (Std) values, in optical crossbar structures using the noise-aware training method. **e** The losses at the final epoch as a function of noise level using noise-aware and noise-unaware training methods.

noise-induced calculation errors are more substantial with larger noise Std values, and accumulate and increase with increasing propagation depth. In particular, errors substantially increase after the fourth processing unit under all Std values, accounting for rapidly increasing validation losses using the noise-unaware training method and highlighting the importance of utilizing noise-aware training for deep ONE architecture.

We further highlight the advantages of the ONE architecture implemented using reconfigurable DONN and XBAR structures in comparison with electronic hardware. The ONE architecture belongs to data-driven approaches, such as FNO, featuring orders of magnitude acceleration compared to typical PDE solvers. Although data-driven approaches require additional training steps, the training is a one-time effort. In contrast, inference is performed over time and accounts for >90% of computing and energy resources[43]. Hence, the benefit of employing data-driven approaches for inference to speed up the process of solving PDEs is overwhelming. Further, for inference, optical system implementations are advantageous over electronic systems in terms of energy consumption and throughput, because optical systems have nearly zero energy consumption (e.g., our liquid-crystal-based SLMs) with frozen weights during inference and the inference computation can be done in a highly parallel single-shot manner. Specifically, we performed quantitative scaling analyses of energy consumption and throughput of the optically implemented ONE architecture and electronically implemented FNO. The FNO model has shown the best prediction and speed performance over other models, as shown in Fig. 2c and ref. 26.

We assume the 2D data has a size of $N \times N$. For linear operations, the energy-intensive ones in the ONE architecture and FNO include Fourier and inverse Fourier transform and matrix multiplications. Both involve complex-valued multiply-accumulate (MAC) operations. Hence, if Fourier and inverse Fourier transform are processed through fast Fourier transform in the FNO, the MAC energy consumption is proportional $(N^2 \log N)E_{MAC}$, where $E_{MAC}$ is the energy consumption for each MAC operation. Further, the energy consumption for processing matrix multiplications is proportional to $N^3 E_{MAC}$. Hence, the total energy consumption of the electronically implemented FNO architecture for linear operations is proportional to $(N^2 \log N + N^3)E_{MAC}$ or $O(N^2 \log N + N^3)$. In stark contrast, the energy consumption of these two linear operations in the optically implemented ONE architecture can be nearly negligible and independent of $N$ or $O(1)$, highlighting the advantages of optical systems to reduce energy consumption in both Fourier-space and real-space processing. Further, in the optical implementation, the energy consumption of the light source scales with the system size to account for detector signal-to-noise ratios. Specifically, if we assume the detector array has the same $N \times N$ size as the 2D data, $E_{det}$ represents the minimum energy needed for each pixel on the detector array, $T_{mod}$ denotes the worst-case transmission in modulators, and $\eta_{fan}$ is the efficiency of the fan-out optics, the light source energy consumption is proportional to $N^2 \eta_{fan} T_{mod} E_{det}$ or $O(N^2)$. Practically, when the system scales up, the needed light source power can be achieved by combining multiple low-power sources if a single high-power source is not accessible. For nonlinear operations, the energy consumption of the ONE architecture and FNO is proportional to $N^2 E_{e,nl}$ and $N^2 E_{o,nl}$, where $E_{e,nl}$ ($E_{o,nl}$) is the energy consumption for processing each element in the electronic (optical) implementation. Hence, considering energy consumptions of linear and nonlinear operations and light sources, the optically implemented ONE architecture has a scaling advantage over the electronically implemented FNO.

Further, the throughput analysis is similar to energy consumption analysis. The system throughput is determined by how fast data can be fed or read out, quantified using input/output (I/O) latency time $t_{I/O}$, and how fast data can be processed, quantified using processing time ($t_{pro}$). In electronic systems, $t_{pro}$ is proportional to $(N^2 \log N + N^3)t_{MAC}$, where $t_{MAC}$ is the time taken to calculate one MAC. In contrast, in

optical systems, $t_{pro}$ is independent of $N$ because of its single-shot processing. For $t_{I/O}$, we consider the worst serial I/O for optical systems, and $t_{I/O}$ is proportional to $N^2 t_{se}$, where $t_{se}$ is the time taken to feed or read out one element from $N \times N$ data. In contrast, we consider the best scenario for electronic systems, and $t_{I/O}$ is independent of $N$. Hence, even comparing the worst case of optical systems and the best case of electronic systems, optical systems have a scaling advantage over electronic systems in terms of throughput because $t_{pro}$ is dominant over $t_{I/O}$ with large $N$.

In addition to 2D data, the ONE architecture can further expand to process higher-dimensional data because higher-dimensional Fourier and inverse Fourier transform and tensor-tensor multiplications can be decomposed into a series of sub-calculations on 2D (inverse) Fourier transform and MVM. The unique advantage of optical systems is to perform all these 2D sub-calculations in parallel through multiplexing, including frequency or wavelength multiplexing[44], path multiplexing (i.e., multiple parallel ONE architectures)[45], polarization multiplexing[46], etc. Assuming the number of multiplexing channels is $M$, the overall latency time is not changed, the throughput is multiplied by $M$, and the energy consumption also scales with $M$.

In our current optical hardware implementation, we can estimate the data processing rate ($R_{pro}$) based on the light propagation time between components, which is ≈1 ns considering the distance between SLMs in the experimental setup shown in Fig. 5a. Assuming 8-bit data, $R_{pro} \approx N^2$ GB/s. On the other hand, the I/O rate of feeding data into the system ($R_{in}$) is dependent on the refresh rate of liquid crystal SLMs, which is 60 Hz and is limited by intrinsically slow liquid crystal response and electrical driving circuits, including digital-to-analog (DAC) converters and addressing mechanisms. Similarly, the I/O rate of reading out data ($R_{out}$) is dependent on the frame rate of the camera, which is 34.8 frames per second and is limited by electrical read-out circuits, such as analog-to-digital converters (ADC). Hence, $R_{in} = 60 \times N^2$ B/s and $R_{out} = 34.8 \times N^2$ B/s, which are much smaller than $R_{pro}$. The I/O rates in current hardware cannot fully unleash the potential of optical computing systems. There are multiple ways to improve I/O rates. For example, to improve $R_{in}$, we can employ SLMs based on intrinsically fast modulation mechanisms, such as a >GHz SLM based on the electro-optic Pockels effect in organic molecules[47]. Meanwhile, we can create an array of fast DACs (e.g., >10 GSamples/s)[31] to match SLM speed and drive each SLM pixel in parallel. A similar array of fast ADCs[31] can be utilized to improve $R_{out}$ in the camera. Further, the improved I/O circuits in optical systems can be seamlessly integrated with high-speed I/O interfaces in state-of-the-art graphic processing unit architecture to fully unleash the high-speed processing advantage in optical systems.

Compared to other metasurface-based analog computing systems[48], our ONE architecture that combines Fourier-space and real-space processing is distinct and unique, and the demonstrated application of solving spatiotemporal 2D PDEs has not been realized in metasurface systems before. Further, the large-scale reconfigurability with individual pixel control of employed commercial diffractive SLMs is crucial in our current optical experimental implementation of the ONE architecture. Such reconfigurability and scalability enable the reuse of limited hardware to achieve deep ONE architectures for complex tasks and the adaptability of the same system for different tasks, significantly reducing the hardware cost and offering the versatility and generalizability of systems. In addition, reconfigurability enables the mitigation of model-hardware discrepancies due to different reasons, which is also important for system scale-up. However, it is still challenging to manufacture large-scale metasurfaces with similar versatile reconfigurability in commercial SLMs. Hence, metasurface technology has not been mature enough to be deployed in our ONE architecture. Indeed, metasurfaces have the potential to achieve a smaller system footprint because their subwavelength components can reduce not only the device footprint but also the propagation

distance between components. Several efforts have demonstrated small-scale reconfigurability by hybridizing metasurfaces with liquid crystals[49], indium tin oxide[50], and electro-optic materials[47]. Future advancement of metasurfaces can offer a promising high-performance implementation solution for the ONE architecture.

## Discussion

We have demonstrated the ONE architecture and validated its performance in solving a broad range of PDEs in diverse scientific domains. The ONE architecture is versatile and can be further modified to reduce the interface and connection between DONN and optical XBAR structures and facilitate the hardware implementation of the whole system. Further, in a whole system, active learning and noise-aware training can be incorporated to mitigate the discrepancy between models and practical systems for accurate deployment. Moreover, in addition to solving PDEs, the ONE architecture can be tailored to accelerate ML models for other regression problems.

## Methods

### DONN diffraction model

The diffraction impulse function $h(x, y)$ was described using the Fresnel equation as

$$h(x, y) = \frac{e^{ikz}}{i\lambda z} e^{\frac{ik}{2z}(x^2 + y^2)}, \tag{16}$$

where $\lambda$ is the wavelength, $k = 2\pi/\lambda$ is the free-space wavenumber, $(x, y)$ are positions within a plane perpendicular to the wave propagation direction, $z$ is the distance along the propagation direction, and i is the imaginary unit. The 2D Fourier transform was directly performed on $h(x, y)$ for model training and evaluation. To match the experimental setup as described below, $h(x, y)$ was first discretized with respect to a defined rectangular mesh grid in the convolution calculation and then converted into the Fourier space through 2D Fourier transform[21].

### The operation mechanism of optical XBAR structures

Supplementary Fig. 1a shows the detailed schematic of an integrated photonic XBAR structure. Specifically, the element values of a $n \times 1$ input vector $\mathbf{v}$ are represented by the intensities of light at input waveguides, $\{I_1, I_2, I_3, \ldots, I_n\}$, which can be implemented by modulating an equally distributed laser intensity through a $n \times 1$ array of electro-optic modulators (red squares in Supplementary Fig. 1a) at input waveguides. The light on each row waveguide is then equally distributed to the column waveguides connected to that row waveguide and modulated through an electro-optic modulator on the coupled curved waveguide (yellow squares in Supplementary Fig. 1a). The element values of a $m \times n$ matrix $M$ are represented by the transmittance of modulators on curved waveguides, $\{T_{ij}\}, i \in [1, m], j \in [1, n]$. At the end of each column waveguide, a photodetector collects all light intensity passing through the column waveguide. The obtained photocurrents or photovoltages of a $m \times 1$ photodetector array represent the summation of multiplied input vector light intensity and matrix modulator transmittance, and the element values of output vector $\mathbf{o}$, $O_j = \sum_{s=1}^{n} T_{js} I_s, j \in [1, m]$. Hence, this integrated photonic XBAR structure can implement MVM in the optical domain.

Similarly, Supplementary Fig. 1b shows the detailed schematic of a free-space optical XBAR structure. Specifically, the element values of a $n \times 1$ input vector $\mathbf{v}$ are represented by the intensities of light, $\{I_1, I_2, I_3, \ldots, I_n\}$, which is implemented through a $n \times 1$ array of free-space vector SLM. The output light is broadcast to a $m \times n$ array of matrix SLM through lenses so that the light distribution from vector SLM is identical at each column of matrix SLM. The element values of a $m \times n$ matrix $M$ are represented by the transmittance of matrix SLM, $\{T_{ij}\}, i \in [1, m], j \in [1, n]$. Lenses are then used to focus the output light from each modulator on the same column of matrix SLM to a

photodetector. The readings from a $m \times 1$ photodetector array represent the element values of output vector $\mathbf{o}$, $O_j = \sum_{s=1}^{n} T_{js} I_s, j \in [1, m]$. Hence, this free-space optical XBAR structure can also implement MVM in the optical domain.

### ONE architecture model

The ONE architecture model was constructed with two main modules – the DONN module processing data in the Fourier space and the optical XBAR module processing linear operations. The mathematical operations in DONN and optical XBAR structures have been described in the first two Methods subsections and their accurate models have been implemented in our prior works, closely matching experimental results[21,42]. Briefly, the DONN module was modeled by combining the Fresnel free-space diffraction with phase-only spatial light modulation in a range of $[0, 2\pi]$ in the model and coupled spatial light modulation as shown in Supplementary Fig. 4; the optical XBAR module was represented as matrix multiplication incorporating measurement noise. The nonlinear function was tanh. The whole model was implemented under the PyTorch 1.12 framework with graphics processing unit (GPU)-accelerated parallel computation and gradient backpropagation for training. The GPU used in this work was an Nvidia RTX 6000 card.

### Darcy flow equation dataset and training

A 2D Darcy flow equation on the unit box was employed. The corresponding PDE is a second-order, linear, elliptic PDE as

$$-\nabla \cdot (k(x, y)\nabla u(x, y)) = f(x, y), \ x \in (0, 1), y \in (0, 1), \tag{17}$$

$$u(x) = 0, \ x \in \partial(0, 1), y \in \partial(0, 1) \tag{18}$$

with a Dirichlet boundary condition. We used the Darcy flow dataset from the existing dataset in ref. 26 with a boundary condition $u(x, y) = 0$ on domain edges. The coefficient $k(x, y)$ was generated based on a specific distribution with the value 12 for positive inputs and 3 for negative inputs. The forcing term was fixed at $f(x, y) = 1$. The solution $u(x, y)$ was computed using a second-order finite difference method on a $421 \times 421$ grid, and other resolutions were obtained with downsampling. We used a 10 : 1 ratio for the numbers of data in the training set and validation set, respectively. The model was trained with a total of 600 epochs and a batch size of 40. The learning rate was 0.1 for the trainable parameters in DONNs and 0.001 for all other trainable parameters with the Adam optimizer.

### Magnetostatic Poisson's equation dataset and training

The demagnetizing field $\mathbf{H}$ originates from the magnetization within the material itself, which can be calculated as the convolution of $\mathbf{M}$ with the demagnetization tensor $\mathbf{N}$ as

$$\mathbf{H}(\mathbf{r}) = \int \mathbf{N}(\mathbf{r} - \mathbf{r}')\mathbf{M}(\mathbf{r}')d\mathbf{r}'. \tag{19}$$

This convolution was computed through Fourier space representations of fields. Specifically, to create the dataset, we utilized the MagneX solver[51] to simulate the time evolution of magnetization in a thin magnetic film with dimensions of $500 \times 125 \times 3.125$ nm. The modeling incorporated both demagnetization and exchange interactions. Initially, we relaxed the magnetic field into a stable S-state before subjecting the system to varying external magnetic fields in different scenarios. We uniformly sampled 8 bias $\mathbf{H}$ fields in the $x$ and $y$ directions, each with a magnitude of 19,872 A/m. The system evolved for 1 ns, during which we collected paired data of $\mathbf{M}$ and $\mathbf{H}$ fields. Each field was represented by three channels corresponding to the field components in $x$, $y$, and $z$ directions. The dataset was divided into training and testing sets with an 8 : 2 ratio. The training was conducted

over 500 epochs with a batch size of 128. The learning rate was set to 1.0 for the trainable parameters in DONNs and 0.001 for all other trainable parameters with the Adam optimizer.

## Navier-Stokes equation dataset and training

A 2D Navier-Stokes equation for a viscous, incompressible fluid in vorticity form on the unit torus was used to generate spatiotemporal data for training the ONE architecture. Specifically, the PDEs are

$$\partial_t w(x,y,t) + u(x,y,t) \cdot \nabla w(x,y,t) = \nu \Delta w(x,y,t) + f(x,y), \; x \in (0,1), y \in (0,1), t \in (0,T] \tag{20}$$

$$\nabla \cdot u(x,y,t) = 0, x \in (0,1), \; y \in (0,1), \; t \in (0,T] \tag{21}$$

$$w(x,y,0) = w_0(x,y), \; x \in (0,1), y \in (0,1), \tag{22}$$

where $w_0(x,y)$ is the initial vorticity and boundary conditions were used. We utilized the existing dataset with the viscosity coefficient $\nu = 10^{-3}$ from ref. [26] for training and inference. The samples in the dataset were recorded with a time step of $10^{-4}$ s. We used 1000 data as the training set and 100 data as the validation set. We trained the ONE architecture model with the first 10 vorticity fields ($w(x,y,t)$) to predict the time evolution of the next 10 vorticity fields. The model was trained with a total of 600 epochs and a batch size of 40. The learning rate was 0.1 for the trainable parameters in DONNs and 0.001 for all other trainable parameters with the Adam optimizer.

## Maxwell's equations dataset and training

We employed commercial Ansys Lumerical finite-difference-time-domain simulation software to generate an electric field dataset by solving Maxwell's equations in dielectric metasurfaces. Specifically, the dielectric metasurface had a periodic pattern and we used four silicon cylindrical rods as the unit cell and periodic boundary condition. Data were generated by randomly selecting the radii of four cylindrical rods. The radius was chosen from 39.5 μm to 44.5 μm with a step of 0.25 μm. The simulation time was set as 300,000 fs. We generated a total of 1200 data and used 1000 as the training set and the rest 200 as the validation set. The model was trained in an auto-regressive style for the $E_x$ component processing. The $E_x$ field data between 29,502 fs to 30,977 fs was fed into the model to predict the next 1,475 fs (10 steps), 2950 fs (20 steps), and 4425 fs (30 steps) $E_x$ field data. The model was trained with a total of 500 epochs and a batch size of 20. The learning rate was 0.1 for the trainable parameters in DONNs and 0.001 for all other trainable parameters with the Adam optimizer.

## Multiphysics dataset and training

We employed commercial COMSOL Multiphysics finite-element simulation software to generate a temperature profile dataset by solving coupled electric current and heat transfer PDEs in an electrical heating circuit[52]. Specifically, the circuit contained a serpentine-shaped Nichrome resistive layer with 10 μm thick and 5 mm wide on top of a glass plate. A silver contact pad with a dimension 10 mm × 10 mm × 10 μm was attached at each end. The deposited side of the glass plate was in contact with the surrounding air at 293.15 K and the back side was in contact with the heated fluid at 353 K. Two coupled physics modules, electrical current in layered shells and heat transfer in layered shells, were used in COMSOL simulations. The input voltage pulse height was set from 5 to 25 V with a step of 1 V and the pulse width was set from 20 to 60 s with a step of 1 s. The simulation time range was from 0 to 110 s. We generated a total number of 861 data and divided the data into training and testing set with the splitting ratio of 8: 2. The ONE architecture took the electric current layer data as the input spatiotemporal data and the input voltage pulse information was fed into the physics parameter data processing branch to predict

temperature field data at 55 s. The model was trained with a total of 100 epochs and a batch size of 40. The learning rate for the trainable parameters in DONNs was 0.1 and the learning rate for all other trainable parameters was 0.001 with the Adam optimizer.

## DONN experimental setup and alignment

The photo and schematic diagram of the DONN experimental setup are displayed in Fig. 5a. The laser diode with a center wavelength of 532 nm and 4.5 mW power (CPS532 from Thorlabs, Inc.) was used as a source. The distance between SLMs and between the last SLM and camera was set as 25.4 cm. The polarizers and half-wave plates before and after each SLM were configured so that each SLM operated with a strong modulation of the transmitted electric field phase (phase mode) together with a moderate modulation of light amplitude. The experimentally measured amplitude and phase modulation responses of three SLMs are shown in Supplementary Fig. 4. All transmissive SLMs are the LC 2012 model from HOLOEYE Photonics AG with a refresh rate of 60 Hz. The analog-to-digital converter has 8-bit precision for liquid crystal driving voltage, so that the gray level of SLMs is from 0 to 255. The pixel size of SLMs is 36 μm × 36 μm. The output data was captured on a CMOS camera with a frame rate of 34.8 frames per second (CS165MU1 from Thorlabs, Inc.).

We aligned the DONN setup by loading standard images on SLMs and comparing experimental results with simulation. Specifically, as shown in Supplementary Fig. 5a, standard Gaussian images, which were centered with a peak at 255 gray level and with a standard deviation of 6 pixels, were loaded in the input SLM and two diffractive SLMs. Supplementary Fig. 5b displays the simulation pattern for the perfectly aligned setup. During the alignment process, loaded images were moved up, down, left, and right pixel-by-pixel to match the captured images by the camera with the simulation pattern. Supplementary Fig. 5c displays the matched experimental diffraction pattern when the optical setup was aligned, while Supplementary Fig. 5d shows misaligned patterns when there was five-pixel misalignment in vertical and horizontal directions, respectively.

## DONN experimental training with reparameterization

The discrete look-up tables of device responses shown in Supplementary Fig. 4 break the gradient backpropagation in the ML training process in PyTorch. To solve this challenge, we utilized a differentiable reparameterization Gumbel-softmax technique, which was first introduced in ref. [53] and demonstrated in our prior work[21]. Specifically, continuous noise from the Gumbel distribution was added to the discrete distribution. The argmax function was then used to find the optimized sample. The training problem after this Gumbel-argmax process is mathematically equivalent to the original training problem under one-hot representation[53]. Since the argmax function still breaks the gradient chain, it was replaced with the softmax function to enable differentiability. Hence, this Gumbel-softmax technique, which is also available in PyTorch, offers continuous and differentiable approximation to discrete distributions and the gradient can backpropagate to reduce the loss function.

## DONN experimental gray-level encoding

The global minimum and maximum values in input 2D data were denoted as $d_{\min}$ and $d_{\max}$. A gray level range from 130 to 255 in the input encoder SLM was selected for a relatively large amplitude modulation range to have enough contrast. Hence, any value $d$ in the input 2D data was converted into a gray level through a linear mapping as

$$\text{int}\left(\frac{(255-130)(d-d_{\min})}{d_{\max}-d_{\min}}+130\right), \tag{23}$$

where the int( $\cdot$ ) operation rounded the expression to the nearest integer since the SLM gray level must be an integer.

## Post-processing DONN experimental results
A simple, lightweight CNN was trained to map experimental results to model results. This simple CNN consists of four convolutional layers with ReLU activations. It takes a single-channel input and processes it through layers with increasing ($1 \rightarrow 16 \rightarrow 64$) and decreasing ($64 \rightarrow 16 \rightarrow 1$) channel dimensions, preserving spatial dimensions with $3 \times 3$ kernels and padding. The final layer outputs a single-channel result.

## Optical XBAR noise
The MVM results from an optical XBAR structure were uniformly randomly generated in a range of $-15$ to $15$, which was the value range in the ONE architecture for solving the Darcy flow equation. The expected number $o$ was then added with a randomly generated noise from a Gaussian distribution with a zero average and varying standard deviation. The noise-dressed number $\bar{o}$ was used in ONE architecture calculations. Under different noise standard deviation levels, Supplementary Fig. 9a demonstrates $\bar{o}$ with respect to $o$ and Supplementary Fig. 9b displays histograms of $\bar{o} - o$.

## Data availability
The source data generated in this study and the datasets and saved models for running codes are publicly available at ref. 54.

## Code availability
The codes that support the plots within this paper and other findings of this study are publicly available at https://github.com/GaoUtahLab/ONE_PDE_public. The Zenodo version is available at ref. 55.

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

## Acknowledgements

R.C., C.Y., and W.G. acknowledge support from the National Science Foundation through Grants No. 2235276 (W.G.), No. 2316627 (W.G.), and No. 2428520 (C.Y. and W.G.). M.L., J.F., and W.G. also acknowledge support from the University of Utah start-up fund. Y.T., Z.Y., and A.N. were supported by the US Department of Energy, Office of Science, Office of Advanced Scientific Computing Research, the Microelectronics Science Research Center Projects for Energy Efficiency and Extreme Environments, under contract no. DE-AC02-05-CH11231 (Nanoscale hybrids: a new paradigm for energy-efficient optoelectronics, A.N. and Z.Y.). Y.T. and Z.Y. were supported by Laboratory Directed Research and Development (LDRD) funding from Berkeley Lab, provided by the Director, Office of Science, of the U.S. Department of Energy under Contract No. DE-AC02-05CH11231 (Z.Y.). This research used resources of the National Energy Research Scientific Computing Center (NERSC), a DOE Office of Science User Facility supported by the Office of Science of the U.S. Department of Energy under Contract No. DE-AC02-05CH11231 (Z.Y.) and under NERSC GenAI award under No. DDR-ERCAP0030541 (Y.T. and Z.Y.).

## Author contributions

Y.T. and W.G. conceived the idea, and W.G. supervised the project. Y.T. constructed models and performed machine learning calculations with the help of M.L., J.F., and C.Y., and under the support of A.N., Z.Y., and W.G. R.C. constructed an optical experimental setup, performed experiments, and performed numerical calculations under the support and supervision of W.G.

## Competing interests

The authors declare no competing interests.
