## [Transparent Peer Review file · Nature Communications]

Optical Neural Engine for Solving Scientific Partial Differential Equations

Corresponding Author: Dr Weilu Gao

Version 0:

Reviewer comments:

Reviewer #4

(Remarks to the Author)

In this manuscript, Tang et al. present a photonic platform called optical neural engine. I think this work is novel and very interesting. Hence, I would like to recommend the publication of this manuscript on Nature Communications. Below are a few additional minor comments:

- (1) What nonlinear functions are used in the demonstrated architecture (Fig. 1a)? Please discuss about it.
- (2) In Fig. 3e, it looks like the relative error figure shows a few sharp lines while there are no such lines in the absolute error. Can authors comment on the reason?
- (3) In the third paragraph of page 14, the authors mentioned the multiplexing can be utilized to handle high-dimensional problems. Adding a few references would be beneficial for the community for the future works.
- (4) In the first paragraph of page 6, the used symbol of big O notation has different format from those in the experimental demonstration section. Please revise it.
- (5) What's the output power of laser diode in the experiment?

(Remarks on code availability)

I have downloaded the codes and check them. The codes are well organized and appears working.

Reviewer #5

(Remarks to the Author)

This paper introduces and experimentally demonstrates an optical neural network hardware architecture called ONE, consisting of a combination of diffractive optical neural networks and fully connected crossbar optical neural networks, for the solution to scientific problems governed by partial differential equations. The paper provides a compelling argument for the utility of their hardware architecture as an alternative to digital approaches and reasonably convincing experimental data backing their claims. The paper is well-written, and the technical level is appropriate for the journal. However, there are a few concerns. I recommend minor revisions addressing the following comments.

1. The claim in the abstract "Although optical systems offer high-throughput and energy efficient ML hardware, there is no demonstration of utilizing them for solving PDEs" and related claims in the text are wrong. Please see the following works. I would recommend the authors amend their claims.

Y. Zhao et al. "Real-Time FJ/MAC PDE Solvers via Tensorized, Back-Propagation-Free Optical PINN Training" arXiv:2401.00413 [cs.LG], 2023. <https://arxiv.org/abs/2401.00413>.

Y. Zhao et al. "Experimental Demonstration of an Optical Neural PDE Solver via On-Chip PINN Training" arXiv:2501.00742 [cs.LG], 2025. <https://arxiv.org/abs/2501.00742>.

2. "Specifically, we performed quantitative scaling analyses of energy consumption and throughput of the optically implemented ONE architecture and electronically implemented FNO, with the latter having shown the best prediction and speed performance over other models"

Is "latter" a typo?

3. The energy scaling analysis on page 14 does not account for meeting the photodetector/camera minimum signal-to-noise ratio which may be at risk for very large diffractive optical neural network systems without also scaling laser power. This should account for optical fan-out, any worst-case insertion (transmissive) losses, and any amplification that may need to be done with increased depth to remain within dynamic range of the SLM. Can the authors comment on their expectations for these practical scaling considerations?

4. I would recommend the authors perform an additional baseline comparing the performance of training only the CNN from Fig. 5 on the Darcy Flow and Navier Stokes datasets, in comparison to the current case that uses the experimental hardware in tandem with the postprocessing CNN. If the CNN has similar performance on its own, it will seem pointless to adopt the tandem experimental system. Furthermore, I would recommend the authors amend their energy/speed analysis to account for any digital post-processing that needs to be done. If the authors disagree, alternatively, please explain how experimental systems much larger than theirs will not also require digital error corrections.

5. Can the authors clarify whether the full ONE architecture (DONN + XBAR) was implemented in their experiments of Fig. 5, or only the DONN piece of the architecture? It appears that this is the case as mentioned in Methods, but it was not completely clear in the main text. Furthermore, can the authors clearly label simulated results and experimental results in all figures, particularly Fig. 5? The authors place most emphasis in this paper on simulations of ONE performance (which, mathematically speaking, is essentially just a 2D FNO architecture) rather than experiments; this fact should be made clear to the reader.

(Remarks on code availability)

The code is of high quality for reproducing the inference results of the authors' model.

Version 1:

Reviewer comments:

Reviewer #4

(Remarks to the Author)

The author has answered all my questions very clearly. I have no other comments. I recommend this paper to be published in Nature Communications

(Remarks on code availability)

The code is correct. I have no comments on it.

Reviewer #5

(Remarks to the Author)

The authors have addressed and satisfied all reviewer comments. The work is significant and impressive. I recommend publication with no further changes.

(Remarks on code availability)

The code is of high-quality to reproduce the authors' model inference results.

We thank all referees for their careful review of our manuscript and thoughtful comments. Below we address each of the questions/comments in detail:

Response to Reviewer #4

Reviewer #4's comment #1: What nonlinear functions are used in the demonstrated architecture (Fig. 1a)? Please discuss about it.

Response to comment #1: The nonlinear activation function we used is the *tanh* function. **In the revised main text, we have included this information. Detailed changes are:**

Location: Main text, Line 778

Changed texts in red color: The nonlinear function was tanh.

Reviewer #4's comment #2: In Fig. 3e, it looks like the relative error figure shows a few sharp lines while there are no such lines in the absolute error. Can authors comment on the reason?

Response to comment #2: We thank Reviewer #4 for this comment. The relative errors shown in Fig. 3e are defined as $\left| \frac{\text{Predicted Value} - \text{Ground Truth}}{\text{Ground Truth}} \right| = \left| \frac{\text{Absolute Error}}{\text{Ground Truth}} \right|$, whereas the absolute errors are defined as the absolute value difference between predicted values and ground truth values. Hence, the sharp lines originate from the division of small ground truth values.

In the revised main text, we have clarified their differences when these two quantities are first referenced. Detailed changes are:

Location: Main text, Line 257 - 259

Changed texts in red color: the absolute error between the ground truth and prediction, and the relative error defined as the ratio of the absolute error over the ground truth

Location: Main text, Line 366 - 369

Changed texts in red color: Note that the sharp lines in the relative error plot of Fig. 3e is due to the division of small ground truth values.

Reviewer #4's comment #3: In the third paragraph of page 14, the authors mentioned the multiplexing can be utilized to handle high-dimensional problems. Adding a few references would be beneficial for the community for the future works.

Response to comment #3: We thank Reviewer #4 for this comment. **In the revised manuscript, we have added the following references about different multiplexing methods:**

- (1) *Wavelength multiplexing*: Duan, Zhengyang, Chen, Hang and Lin, Xing. "Optical multi-task learning using multi-wavelength diffractive deep neural networks" *Nanophotonics*, vol. 12, no. 5, 2023, pp. 893-903. <https://doi.org/10.1515/nanoph-2022-0615>
- (2) *Path multiplexing*: Li, Y., Chen, R., Sensale-Rodriguez, B. et al. Real-time multi-task diffractive deep neural networks via hardware-software co-design. *Sci Rep* 11, 11013 (2021). <https://doi.org/10.1038/s41598-021-90221-7>.
- (3) *Polarization multiplexing*: Luo, X., Hu, Y., Ou, X. et al. Metasurface-enabled on-chip multiplexed diffractive neural networks in the visible. *Light Sci Appl* 11, 158 (2022). <https://doi.org/10.1038/s41377-022-00844-2>

Detailed changes are:

Location: Main text, Line 667 – 668

References: [44], [45], [46]

Reviewer #4's comment #4: In the first paragraph of page 6, the used symbol of big O notation has different format from those in the experimental demonstration section. Please revise it.

Response to comment #4: In the revised main text, we have revised the symbol notation. Detailed changes are:

Location: Main text, Line 255
Changed texts in red color: $O(1)$

Reviewer #4's comment #5: What's the output power of laser diode in the experiment?

Response to comment #5: The output power of the laser diode used in the experiment is 4.5 mW. In the revised main text, we have included this information. Detailed changes are:

Location: Main text, Line 872
Changed texts in red color: The laser diode with a center wavelength of 532 nm and 4.5 mW power (CPS532 from Thorlabs, Inc.) was used as a source.

Response to Reviewer #5

Reviewer #5's comment #1: The claim in the abstract “Although optical systems offer high-throughput and energy efficient ML hardware, there is no demonstration of utilizing them for solving PDEs” and related claims in the text are wrong. Please see the following works. I would recommend the authors amend their claims.

Y. Zhao et al. “Real-Time FJ/MAC PDE Solvers via Tensorized, Back-Propagation-Free Optical PINN Training” arXiv:2401.00413 [cs.LG], 2023. <https://arxiv.org/abs/2401.00413>.

Y. Zhao et al. “Experimental Demonstration of an Optical Neural PDE Solver via On-Chip PINN Training” arXiv:2501.00742 [cs.LG], 2025. <https://arxiv.org/abs/2501.00742>.

Response to comment #1: We thank Reviewer #5 for pointing out these prior works. These papers primarily explore the use of optical matrix-vector multipliers (MVMs) within the **conventional physics-informed neural network (PINN) framework**. The first preprint avoids backpropagation by using zeroth-order gradient estimation and proposes to use tensor-train decomposition to reduce the number of optical components in optical MVM hardware. The second preprint presents a small-scale (1×4) experimental setup based on microring resonators to solve a basic heat equation with an analytical solution $(\sin(\pi x)e^{-t})$. In contrast, our work presents a fundamentally different optical neural engine (ONE) architecture that combines diffractive optical neural networks (DONNs) and optical MVMs, moving beyond the traditional PINN framework. Our ONE architecture enables direct optical representation learning for solving complex and coupled multiphysics PDEs with high performance. Furthermore, our experimental demonstrations of DONNs are at a substantially larger scale ($\sim 100 \times 100$), demonstrating the feasible architecture scalability using free-space optics for solving PDEs of greater complexity than those considered in prior works.

In the revised main text, we have amended our claims as “the demonstration of utilizing optical systems for solving PDEs is limited” in the abstract and introduction and cited these prior works. Detailed major changes are:

Location: Main text, Line 31

Changed texts in red color: *their demonstration for solving PDEs is limited.*

Location: Main text, Line 78 – 79

Changed texts in red color: *However, the deployment of optical computing systems is in small scales for basic PDEs with limited performance [24,25].*

Reviewer #5's comment #2: “Specifically, we performed quantitative scaling analyses of energy consumption and throughput of the optically implemented ONE architecture and electronically implemented FNO, with the latter having shown the best prediction and speed performance over other models” Is “latter” a typo?

Response to comment #2: We thank Reviewer #5 for this comment. What we would like to state is: “the Fourier Neural Operator (FNO) model has shown the best prediction and speed performance over other models on electronic hardware, as shown in Fig. 2c and Ref. [26]”.

In the revised main text, we have revised this sentence to make this point clearer. Detailed changes are:

Location: Main text, Line 620 – 621

Changed texts in red color: *The FNO model has shown the best prediction and speed performance over other models on electronic hardware, as shown in Fig. 2c and Ref. [26].*

Reviewer #5's comment #3: The energy scaling analysis on page 14 does not account for meeting the photodetector/camera minimum signal-to-noise ratio which may be at risk for very large diffractive optical neural network systems without also scaling laser power. This should account for optical fan-out, any worst-case insertion (transmissive) losses, and any amplification that may need to be done with increased depth to remain within the dynamic range of the SLM. Can the authors comment on their expectations for these practical scaling considerations?

Response to comment #3: We thank Reviewer #5 for this insightful comment. We agree with Reviewer #5. When the system size of a diffractive optical neural network (DONN) system scales up, the incident laser power/energy also needs to scale up to account for the camera signal-to-noise ratio. Let's assume the 2D data to process has a dimension of $N \times N$, and the camera has the same dimension. If E_{det} represents the minimum energy needed for each pixel on the camera, T_m represents the worst-case transmission in modulators, and η_{fan} is the efficiency of the fan-out optics, the total output laser energy would be $N^2 \eta_{\text{fan}} T_m E_{\text{det}}$. Hence, the scaling law of laser energy (or power) is also dependent on N^2 . As a result, the conclusion that our optically implemented optical neural engine architecture has a scaling advantage by accelerating $O(N^3)$ multiplication-accumulate operations still holds. As a side note, practically, when the system scales up, the needed laser power can be implemented by combining multiple low-power lasers if a single high-power laser is not accessible.

In the revised main text, we have summarized previous discussions. Detailed changes are:

Location: Main text, Line 635 – 643

Changed texts in red color: Further, in the optical implementation, the energy consumption of the light source scales with the system size to account for detector signal-to-noise ratios. Specifically, if we assume the detector array has the same $N \times N$ size as the 2D data, E_{det} represents the minimum energy needed for each pixel on the detector array, T_m denotes the worst-case transmission in modulators, and η_{fan} is the efficiency of the fan-out optics, the light source energy consumption is proportional to $N^2 \eta_{\text{fan}} T_m E_{\text{det}}$, or $O(N^2)$. Practically, when the system scales up, the needed light source power can be achieved by combining multiple low-power sources if a single high-power source is not accessible.

Reviewer #5's comment #4: I would recommend the authors perform an additional baseline comparing the performance of training only the CNN from Fig. 5 on the Darcy Flow and Navier Stokes datasets, in comparison to the current case that uses the experimental hardware in tandem with the postprocessing CNN. If the CNN has similar performance on its own, it will seem pointless to adopt the tandem experimental system. Furthermore, I would recommend the authors amend their energy/speed analysis to account for any digital post-processing that needs to be done. If the authors disagree, alternatively, please explain how experimental systems much larger than theirs will not also require digital error corrections.

Response to comment #4: We thank Reviewer #5 for this constructive comment. Based on Reviewer #5's suggestion, we have performed additional baseline calculations by employing only post-processing CNNs without diffractive optical neural networks in the optical neural engine (ONE) architecture for the Darcy flow and Navier Stokes datasets. Their loss curves are shown in Fig. R1a and R1b, respectively. We can observe that losses for the CNN baseline architecture are very high in both datasets, showing very poor performance. This comparison clearly highlights the necessity and significance of diffractive optical neural networks in the ONE architecture.

Fig. R1: Loss curves for the ONE architecture and CNN baseline architecture (a) for solving the Darcy flow equation under different resolutions and (b) for solving the Navier-Stokes equation.

Regarding the post-processing circuits, we would like to argue that they **could be removed**. First, the reason why we introduce the lightweight post-processing CNN is based on one suggestion of a previous reviewer. The excellent agreement between post-processed experimental results and calculation model results indicates that our experimental results are indeed close to calculation results and can be corrected with relatively **lightweight** CNN. In future large-scale systems, there are multiple ways to reduce the discrepancy between experiments and calculations from both hardware and software perspectives, which are worth our future studies.

From the hardware perspective, a few reasons contributing to the observed discrepancy include laser speckles and detector noise. For example, incorporating moving diffusers in optical systems can remove speckles (e.g., Serge Lowenthal and Denis Joyeux, "Speckle Removal by a Slowly Moving Diffuser Associated with a Motionless Diffuser," J. Opt. Soc. Am. 61, 847-851 (1971)) and employing high-end cameras with high signal-to-noise ratios (e.g., Electron Multiplying Charge-Coupled Device) can improve detection noise performance.

From the software perspective, the observed discrepancy is because the physical model used for training cannot fully capture experimental variations. As already mentioned in the manuscript, this can be mitigated using a **physics-aware training** approach by incorporating loss functions calculated from experimental results for gradient calculations and hardware reconfiguration, as demonstrated in Ref. [20,38,41] of the revised manuscript. Note that this approach is a training process, only requiring one-time or occasional efforts. This will not affect our energy/speed analysis for inference.

In the revised main text, we have summarized previous discussions and added Fig. R1 in the Supplementary Information (SI) document. Detailed changes are:

Location: Main text, Line 551 – 564

Changed texts in red color: In addition, we trained a baseline architecture by replacing DONN kernels in the architecture shown in Fig. 1a with lightweight post-processing CNN models for solving Darcy flow equations under different resolutions and the Navier-Stokes equation (Supplementary Fig. 8). The losses for the CNN baseline architecture are high in both datasets with poor performance when compared to the results obtained from the ONE architecture with DONN kernels, highlighting the necessity and significance of DONN systems and the limited capability of CNN models.

Instead of post-processing, model-experiment discrepancies can be mitigated through the improvement of system hardware and novel training approaches in future implementation. From the hardware perspective, for example, incorporating moving diffusers in optical systems to remove speckles [40] and employing high signal-to-noise ratio cameras can reduce discrepancies. Moreover, from the training perspective, ...

Location: SI

Added: Supplementary Fig. 8

Reviewer #5's comment #5: Can the authors clarify whether the full ONE architecture (DONN + XBAR) was implemented in their experiments of Fig. 5, or only the DONN piece of the architecture? It appears that this is the case as mentioned in Methods, but it was not completely clear in the main text. Furthermore, can the authors clearly label simulated results and experimental results in all figures, particularly Fig. 5? The authors place most emphasis in this paper on simulations of ONE performance (which, mathematically speaking, is essentially just a 2D FNO architecture) rather than experiments; this fact should be made clear to the reader.

Response to comment #5: In Fig. 5, we experimentally demonstrated only the DONN piece of the architecture. However, in our prior work, we have experimentally demonstrated a free-space optical XBAR system (see the paper: Fan, Jichao, Yingheng Tang, and Weilu Gao. "Universal Approach for Calibrating Large-Scale Electronic and Photonic Crossbar Arrays." *Advanced Intelligent Systems* 5.10 (2023): 2300147). The experimental system output is represented by an error histogram of multiplication results (e.g., see Fig. 5a of the previous paper). Hence, in Fig. 5e, we evaluated the performance of the ONE architecture under different histograms to represent different experimental XBAR system performances (Supplementary Information Fig. 7), and the results of Fig. 5e are indeed **experiment-relevant**. In addition, we agree with Reviewer #5 that all figures should be clearly labeled to indicate whether the results are experimental or simulated.

In the revised main text, we have clarified the experimental demonstration and clearly labeled all figures. Detailed changes are:

Location: Main text, Line 576 – 581

Changed texts in red color: Note that we did not experimentally construct an XBAR system for the ONE architecture. Instead, we added random Gaussian noise with zero mean and varying standard deviation (Std) to the values obtained from matrix multiplications in models to represent experimental hardware noise, such as shot noise in photodetectors as shown in our prior experimental demonstration of a free-space XBAR system [42].

Location: Main text, Figure 2 Caption

Changed texts in red color: Simulation results of solving Darcy flow and magnetostatic Poisson's equations.

Location: Main text, Figure 3 Caption

Changed texts in red color: Simulation results of solving time-dependent Navier-Stokes and Maxwell's equations.

Location: Main text, Figure 4 Caption

Changed texts in red color: Simulation results of solving multiphysics PDEs.

Response to Reviewer #4

Reviewer #4's comment #1: The author has answered all my questions very clearly. I have no other comments. I recommend this paper to be published in Nature Communications.

Response to comment #1: We thank the reviewer for the nice comment.

Response to Reviewer #5

Reviewer #5's comment #1: The authors have addressed and satisfied all reviewer comments. The work is significant and impressive. I recommend publication with no further changes.

Response to comment #1: We thank the reviewer for the nice comment.